# Flowering Phenology of Six Seasonal-Flowering Strawberry Cultivars in a Coordinated European Study

Erika Krüger [1,*], Tomasz L. Woznicki [2], Ola M. Heide [3], Krzysztof Kusnierek [2], Rodmar Rivero [3], Agnieszka Masny [4], Iwona Sowik [4], Bastienne Brauksiepe [5], Klaus Eimert [5], Daniela Mott [6], Gianluca Savini [6], Marino Demene [7], Karine Guy [7], Aurélie Petit [7], Beatrice Denoyes [8] and Anita Sønsteby [2]

1   Department of Pomology, Hochschule Geisenheim University, D-65366 Geisenheim, Germany
2   Norwegian Institute of Bioeconomy Research (NIBIO), NO-1431 Ås, Norway
3   Faculty of Environmental Sciences and Natural Resource Management, Norwegian University of Life Sciences, NO-1432 Ås, Norway
4   The National Institute of Horticultural Research (INHORT), PL-96-100 Skierniewice, Poland
5   Department of Molecular Plant Science, Hochschule Geisenheim University, D-65366 Geisenheim, Germany
6   Sant'Orsola Società Cooperativa Agricola, Via Lagorai, IT-38057 Pergine Valsugana, Italy
7   INVENIO, Maison Jeannette, FR-24140 Douville, France
8   Biologie du Fruit et Pathologie, UMR 1332, INRAE, University of Bordeaux, F-33140 Villenave d'Ornon, France
*   Correspondence: erika.krueger@hs-gm.de; Tel.: +49-61-2379-3879

**Abstract:** The flowering phenology of six genetically distant strawberry cultivars ('Candonga®' (ES), 'Clery' (IT), 'Florence' (UK), 'Frida' (NO), 'Gariguette' (FR), and 'Sonata' (NL)) was studied for 3 years in relation to climatic parameters in open-field cultivation at three locations (Norway, Poland, Germany) and in soil-less cultivation at two locations (Italy, and France), covering a distance of 16 degrees of latitude. This proved to be a useful approach for unravelling the climatic adaptation and plasticity of strawberry genotypes and their suitability both for profitable cultivation and as a breeding pedigree. Despite the intercorrelated character of the climatic variables, the observed results highlight the importance of global radiation as a powerful modifying phenological factor in strawberry. Generally, early flower initiation was associated with elevated temperature and global radiation. 'Frida' revealed the highest dependency on global radiation for flower initiation, while 'Sonata' was least affected by temperature and radiation. In general, temperature and global radiation in periods both preceding and following flower initiation had a stronger positive effect on the number of flowers than on crowns, especially under open-field conditions. The influence of these factors was highly variable across the cultivars: 'Clery', 'Florence', and 'Gariguette' were most affected, while 'Frida' was least influenced.

**Keywords:** climate; flower initiation; *Fragaria* x *ananassa*; global radiation; photoperiod; temperature

## 1. Introduction

The physiology of flowering in seasonal-flowering (SF) or June-bearing strawberry cultivars is widely studied and well documented [1–3]. These cultivars are quantitative short-day (SD) plants that initiate flower primordia in late summer and early autumn when the daylength decreases below a critical length of 14–15 h. Not only long days (LD) but also SD with night interruption (low-intensity light rich in far-red given for 3 h in the middle of the dark period) will prevent floral initiation in SF cultivars, thus demonstrating the true photoperiodic nature of flowering control in the strawberry.

However, the SD flowering response varies greatly with temperature [2,4,5]. At temperatures lower than 12–15 °C, most SF cultivars are more or less day neutral and initiate floral primordia even in continuous light. At temperatures ≥18 °C, they are obligatory SD plants, while excessively high temperatures of 27–30 °C suppress flowering

regardless of daylength conditions. The optimum temperature for SD floral induction and initiation as measured by the number of flowers formed or the number of days to anthesis is in the 18–21 °C range, although both the optimum and threshold temperatures can vary considerably among cultivars [3].

In general, cultivars selected and used under high-latitude conditions have particularly high threshold temperatures, and some of these can initiate flower primordia under long-day (LD) conditions even at temperatures as high as 18 to 21 °C [3]. On the other hand, cultivars such as 'Senga Sengana', 'Korona', and 'Elsanta' are unable to initiate floral primordia in LD even at temperatures as low as 9–12 °C [3]. Generally, low night temperatures can, to some extent, compensate for high day temperatures; however, Verheul et al. (2007) [6] found that the extent of flowering in 'Korona' and 'Elsanta' was as closely related to the mean daily temperature as to specific day and night temperatures. Although flower initiation (FI) can take place at temperatures as low as 3 to 5 °C, it is markedly reduced at temperatures below 9 °C [7].

The minimum number of inductive SD cycles required for the induction of flowering in SF strawberries varies between 7 and 14, depending on the genotype, temperature, and length of the photoperiod [2,3]. With continued SD treatment, the number of flowers increases more or less linearly for at least up to 49 days [8]. This increase is mainly due to an increasing number of flower trusses (inflorescences), while the number of flowers per truss decreases with an increasing number of SDs. After transfer to noninductive LD conditions, further floral initiation ceases sharply, whereas the development of existing floral primordia to anthesis is actually advanced by LD [2,5]. Since floral initiation results in crown branching [2], the crown number will vary in parallel with the number of inflorescences. Elevated temperature during the differentiation of floral primordia after the completion of the SD floral induction and initiation period will also generally enhance and advance flowering in SF strawberries. This is apparently an effect of an enhanced growth rate at higher temperatures.

In addition to the photoperiod and temperature, other environmental factors, such as water, mineral nutrients, and radiation, can also influence the flowering process in strawberry. A complication with these factors is, however, that they may influence flowering both directly and indirectly through their effect on growth. Thus, temporary drought and nutrient deficiency stress may cause floral initiation under otherwise noninductive conditions [2,3], whereas at the same time, irrigation and fertilization may increase flowering by producing a larger plant with more potential flowering sites. An increasing nutrient supply from a low level usually increases flowering and yield, while a surplus, especially of nitrogen (N), can inhibit flower formation, although withholding N may not increase flowering and yield [2]. Such conflicting findings may be due to the indirect and promotive effects of nutrition on plant growth that will increase plant size and, thereby, the number of potential flowering sites. On the other hand, a pulse of a generous N supply concurrent with SD exposure will enhance and promote flowering in SF cultivars independent of plant growth [9,10]. The effect was greatest when the N supply started a few days after transition to inductive conditions, while application immediately before transition to SD delayed and reduced flowering. This latter result concurs with the common practical experience that excessive plant lushness caused by generous N fertilization and irrigation during August may delay and reduce flowering.

Similar direct and indirect effects on flowering may also be expected by the variation in daily radiation. On the one hand, the attainment of a satisfactory plant energy level is required for a normal flowering response in general [11]; additional radiation may also increase flowering through the enhancement of plant size, as explained above. Especially when the period of vegetative growth prior to SD induction is marginal as with late planting, the sum of photosynthetic active radiation (PAR) received by the crop prior to SD exposure is likely to have large positive effects on the number of flowers produced [12].

Since flowering in strawberry genotypes is documented to be differently affected by environmental factors, and since climate change will alter temperatures on site, a better

knowledge about cultivar adaptability or resilience to different regional environmental conditions and to future climates is required. Breeders will need such information for a better climate-adapted breeding strategy and growers for the best cultivar choice for their growing conditions. Therefore, a coordinated transnational trial from north to south of Europe was performed to study the plant performance of already-well-established cultivars with a distinct genetic background and contrasting flowering time.

## 2. Materials and Methods

### 2.1. Experimental Sites, Plant Material, and Cultivation

Flower initiation and plant performance were studied in six June-bearing strawberry genotypes in the years 2016–2019. Each year, the experiment started in August and ended after anthesis the next spring. Plants were grown at five locations from north to south of Europe: in Kapp, Norwegian Institute of Bioeconomy Research (NIBIO), Norway (60°40′ N); Skierniewice, National Institute of Horticultural Research (INHORT), Poland (51°95′ N); Geisenheim, Hochschule Geisenheim University (HGU), Germany (49°59′ N); Pergine Valsugana (Sant'Orsola Società Cooperativa Agricola), Italy (46°40′ N); and Douville (Invenio), France (44°85′ N), hereafter named as Norway, Poland, Germany, Italy, and France. As common for the different regions, experiments in Norway, Poland, and Germany were carried out in an open field, whereas in Italy and France, they were performed in polytunnels that were opened sideways from the beginning of anthesis. Details of the respective latitude, altitude, yearly mean temperature, and type of cultivation as well as soil type are given in Table 1. Air temperature ($T_{mean}$, $T_{max}$, and $T_{min}$) and global radiation were recorded at each location. Additional climatic data for the experimental periods are presented in Table A1.

**Table 1.** Geographic latitude and altitude, yearly mean temperature, cultivation conditions, and soil type for the five experimental locations in Europe.

|  | NIBIO Norway | INHORT Poland | HGU Germany | Sant'Orsola Italy | INVENIO France |
|---|---|---|---|---|---|
| Latitude | 60°40′ N | 51°95′ N | 49°59′ N | 46°4′ N | 44°85′ N |
| Altitude (m a.s.l.) | 262 | 252 | 95 | 925 | 145 |
| Yearly mean temperature (°C) [a] | 5.0 | 7.9 | 9.9 | 11.3 | 12.9 |
| Type of cultivation | Open field | Open field | Open field | Plastic tunnel | Plastic tunnel |
| Soil type | Loam | Pseudopodsol with light clay | Sandy loam | Soil-less culture | Soil-less culture |

[a] For the period 1981–2010.

The six June-bearing strawberry genotypes used in this study were selected by their diverse genetic background and known adaptability to different geographic local environmental conditions: 'Candonga®' (ES), 'Clery' (IT), 'Florence' (UK), 'Frida' (NO), 'Gariguette' (FR), and 'Sonata' (NL). Plants of all cultivars were grown in Norway, Poland, Germany, and France, while only 'Clery', 'Frida', 'Gariguette', and 'Sonata' were grown in Italy. All locations propagated their own plants from mother plants purchased from the same nursery the first year (2016). Due to problems with shipping of plants, and receiving them in good condition, all locations established their own mother plants of 'Candonga', 'Florence', 'Frida', and 'Sonata' in autumn 2016 for the further yearly propagation of runner plants. In addition, each location received new mother plants of 'Clery' and 'Gariguette' every year from the same nursery as in 2016. Moreover, Germany received new plants of 'Candonga' to be used as mother plants in 2017. Thus, each year, plug plants were planted on week 32 in Norway, Poland, and Germany in an open field for studying the time of flower initiation in the planting year, and anthesis as well as plant architecture the following spring. In Italy and France, however, the plants remained in the trays until their transfer into a cold store in December. The cold-stored plants were then established in peat bags on week 9 for the upcoming season. Plant protection, fertilization and irrigation in the

open field sites, and fertigation of the plants in peat bags were performed according to local guidelines and recommendations. These parameters were not treated as factors, although such cultivation practices might have an effect on the studied parameters. However, the agronomical practices were performed as similar as possible in the different locations by using a unified protocol.

### 2.2. Flower Initiation and Development

Starting on week 32, plants were sampled randomly at 10-day intervals until mid-October. At each sampling date, the main crown of four plants from each of three replicates grown in each of the experimental sites (i.e., 12 crowns of each cultivar) was dug and stored on glass vials in 70% ethanol at 5 °C until being dissected under a stereo microscope (magnification: 40 to 60-fold). The flowering stages of the apex were scored according to the following nine-stage scale modified after Taylor et al. [13]:

- Stage 1 = vegetative apex with only leaf primordia;
- Stage 2 = apex round and slightly elevated, apical dome above level of developing stipules;
- Stage 3 = Sepal primordia visible in terminal bud;
- Stage 4 = Secondary flower primordia is visible;
- Stage 5 = Petal primordia visible in terminal bud;
- Stage 6 = Whorls of stamen visible in terminal bud, prolonged sepals cover petals;
- Stage 7 = Carpel primordia visible, central region still undifferentiated;
- Stage 8 = Initiation of carpels complete, anthers still green;
- Stage 9 = All flower parts differentiated, anthers yellow.
- Here, we use floral Stage 2 for the day of the year of flower initiation (DOY-FI). DOY-FI was interpolated using the spline smoothing function.

### 2.3. Date of Anthesis and Plant Architecture the Next Spring

The date of anthesis was recorded when 50% of the plants of each replicate and cultivar had one open flower based on two observations per week. In Norway, Poland, and Germany, each plot comprised three randomized blocks with 20 plants of each cultivar as single rows under open-field conditions (1 × 0.3 m), whereas in Italy and France, because of the soil-less culture, the experimental design consisted of double rows with three replications per genotype with 22–24 plants each. At the end of the flowering period, the number of crowns and flowers of five additional plants of each replicate (i.e., 15 plants of each cultivar) was counted after digging the plants out of the soil and separating them into the different plant organs.

### 2.4. Plant Traits and Environmental Factors

In order to evaluate the impact of latitude-related environmental factors on the time of FI and anthesis as well as the number of crowns and flowers per plant, daily temperature ($T_{mean}$, $T_{max}$, $T_{min}$), global radiation, and photoperiod were calculated based on recorded data for all locations and cultivars and expressed as means of the relevant period: (a) for the 3-week period before FI and (b) for the 5-week period after FI. For the 3-week period before anthesis, growing degree days (GDD) were calculated according to the equation $GDD = [(T_{max} + T_{min})/2] - T_{Base}$, where $T_{max}$ and $T_{min}$ are the mean daily maximum and minimum temperature and $T_{base}$ is the threshold temperature below which the plant development is markedly reduced. In the present study, we checked results using 3, 4, 5, and 6 °C as $T_{base}$, and based on the strongest output, only results with 5 °C is presented here. In addition, the mean global radiation was also calculated based on the measured data for the 3-week period before anthesis. The periods were defined by the well-known critical number of inductive SD cycles required for the induction of flowering in SF strawberries [3] and the limiting Norwegian climate condition in autumn after flower initiation (short period between FI and frost) and in spring before anthesis (short and intensive period after snowmelt).

### 2.5. Statistical Analyses

Since plant traits were registered at 10-day intervals, the cubic spline smoothing function within the R 4.2.1 statistical software (R Core Team, 2021) was applied to the registered data after the removal of outlying samples to interpolate apex flowering stages. The root mean square error of the spline function, calculated on the days of the actual registration, averaged at 0.54 (floral dev. stage), which was considered acceptable, taking into account that the resolution of the registrations was 10 days and that the nominal registration step was 1. The date of the actual flower initiation was then extracted from the interpolated data (Figure A1).

To test whether climatic parameters and genotype are affecting plant physiological responses, statistical variance analysis (ANOVA) was used to detect the differences between physiological traits, and simple linear regression evaluated the relationship between the environmental conditions of the different locations and cultivar-specific needs of temperature, global radiation, and photoperiod before and after FI and before anthesis. The analyses were performed using SPSS version 26. Significant differences were calculated using post hoc Tukey's test at a 5% probability level. Principal component analysis (PCA) was performed using the Minitab$^{®}$ Statistical Software program package (Release 19, Minitab Inc., State College, PA, USA). PCA was built using a correlation matrix and R-mode. PC1, PC2, and PC3 represented 80.2% of the total variance. The selection is based on the size of eigenvalues and the Kaiser criterion (>1).

## 3. Results

### 3.1. Flower Initiation

The timing of flower initiation (FI) in the experimental years 2016–2018 was identified when plants showed the first stage of floral transition, Stage 2. This stage was derived for the different locations from a spline smoothing function enveloped over the registered data that included the various floral developmental stages. The obtained results, together with the statistical interpretation, are presented in Figure 1 and Table A2.

The results confirmed that the FI process in strawberry is controlled by photoperiod and temperature (optimum at 14 h and 18 °C, respectively) (Figure 1a). In Norway, the photoperiodic optimum are reached late, while in France, the temperature is above the optimum level during late summer, both resulting in a delayed DOY-FI. Global radiation (Figure 1b) is linked to photoperiod and temperature, but the values are usually fluctuating over shorter time intervals, which was especially emphasized in the results between the years. These fluctuations might modify the predominant effect of temperature and photoperiod, as demonstrated for Norway and Germany in 2016 and 2017, weeks before DOY-FI (cf. Figure 1a,b). It should be noticed, however, that plants were grown in an open field in Norway, Poland, and Germany, while in Italy and France, the plug plants were raised in trays and cultivated in a substrate in an open field; therefore, the obtained results might be also affected by such factors.

The timing of FI differed between the cultivars starting with 'Frida', followed by 'Clery' 5–6 days, and 'Sonata' 7 days later, respectively. The latest cultivars were 'Candonga', followed by 'Florence'. However, the timing of FI was not fully consistent across the locations (Table A2). For example, 'Frida' was the earliest cultivar in all locations except in Norway, where 'Sonata' and 'Clery' initiated flowers a bit earlier. In addition, 'Florence' behaved differently in Norway, initiating flowers before 'Candonga'. Essentially, the highest variability between the dates of FI across the locations was observed in 'Frida' with 21 days, whereas only 13 days' variation was noted for 'Sonata'.

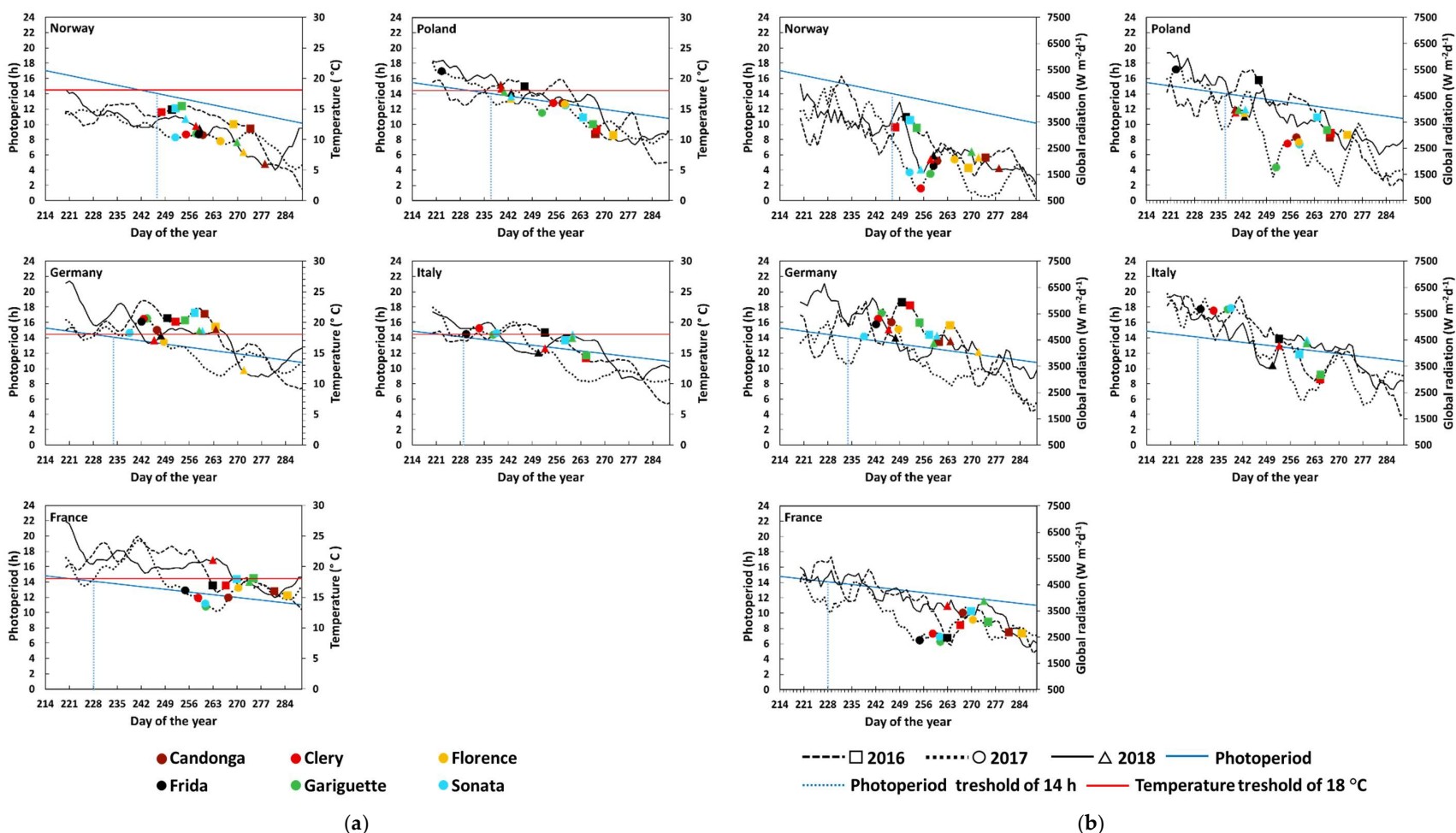

**Figure 1.** Day of flower initiation (floral developmental Stage 2) of six cultivars grown at five locations in Europe as influenced by (**a**) photoperiod, temperature, and (**b**) photoperiod and global radiation during the 3 years of study. The data are presented as a running average of 7 days.

In order to obtain detailed information on the environmental effect on FI in strawberry, the climatic conditions across the locations 3 weeks before DOY-FI are presented as a principal component analysis (PCA) (Figure 2). The score plot of PCA (Figure 2a) reveals the clear separation of the experimental locations and confirms the most distinct climatic conditions in Norway and France. The loading plot (Figure 2b) provides additional information about the relationship between the investigated climatic data and DOY-FI. Variables that are close to each other on the plot denote a strong positive correlation, while a strong negative correlation is present for factors that are symmetrically distant on the loading plot area.

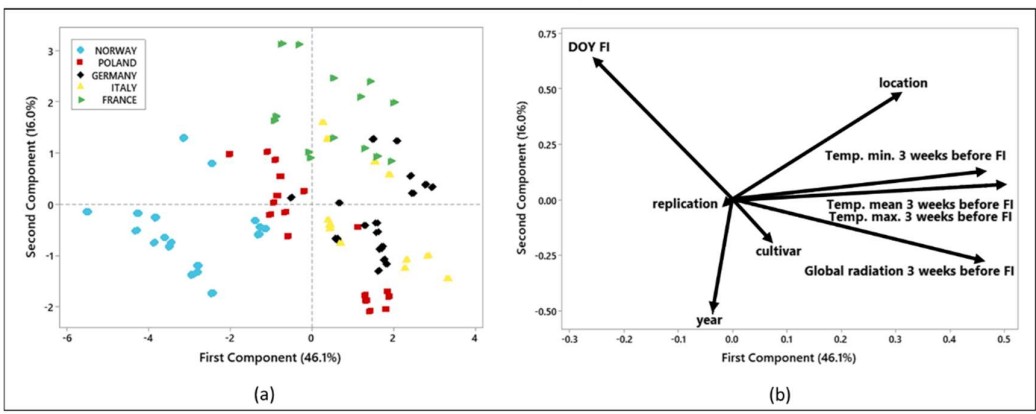

**Figure 2.** Principal component analysis (PCA) based on the correlation matrix of the day of the year for strawberry flower initiation (DOY-FI) and climatic variables during the 3-week period preceding flower initiation. Score plot (**a**) and loading plot (**b**).

When PCA was performed for the most dominant factors ($T_{mean}$ and global radiation 3 weeks before FI), PC1 revealed 41%, PC2 22.9%, and PC3, 16.3% of the total variance, representing 80.2% of the total variance. The highest loading values for PC1 were obtained for environmental factors' global radiation (0.595) and $T_{mean}$ (0.587). PC2 can be characterized by location (loading vector = 0.567) and year (loading vector = −0.558), while PC3 is dominated by cultivar (loading vector = 0.967).

Based on the obtained information for PC1, global radiation and $T_{mean}$ 3 weeks before FI had the strongest relation to DOY-FI, and this is further confirmed by the detailed regression analyses (Figures 3 and A2). The strongest negative relation between global radiation 3 weeks before FI and DOY-FI was observed in Italy, Norway, and Poland, while for $T_{mean}$, high negative $R^2$-values were obtained in a descending order for Poland, Italy, and Norway (Figure A2). The other two locations (Germany and France) revealed little or no effect, probably because of lower variability in climatic conditions in Germany and only 2 years of study for most of the cultivars in France (Figures 2a and A2). The very high $R^2$-values obtained for Italy can be explained by the absence of the late-initiating cultivars 'Candonga' and 'Florence' (Figure A2).

For cultivars, the comparison of temperature and the highly intercorrelated global radiation confirms the slightly dominant effect of the last parameter in relation to DOY-FI (Figure 3a,b). However, temperature is significantly modulating the timing of FI mainly in cultivars that initiated flowers late ('Candonga' and 'Florence'). On the other hand, analysis of the time of FI in 'Frida', which is known for its early FI, revealed the highest dependency on global radiation (the largest differences both between the years in the same location and across the locations). In contrast, 'Sonata 'seems to be less dependent on radiation and temperature conditions for FI. This cultivar initiated flowers almost uniformly across the experimental locations, proving its plasticity and adaptation to a range of growth conditions.

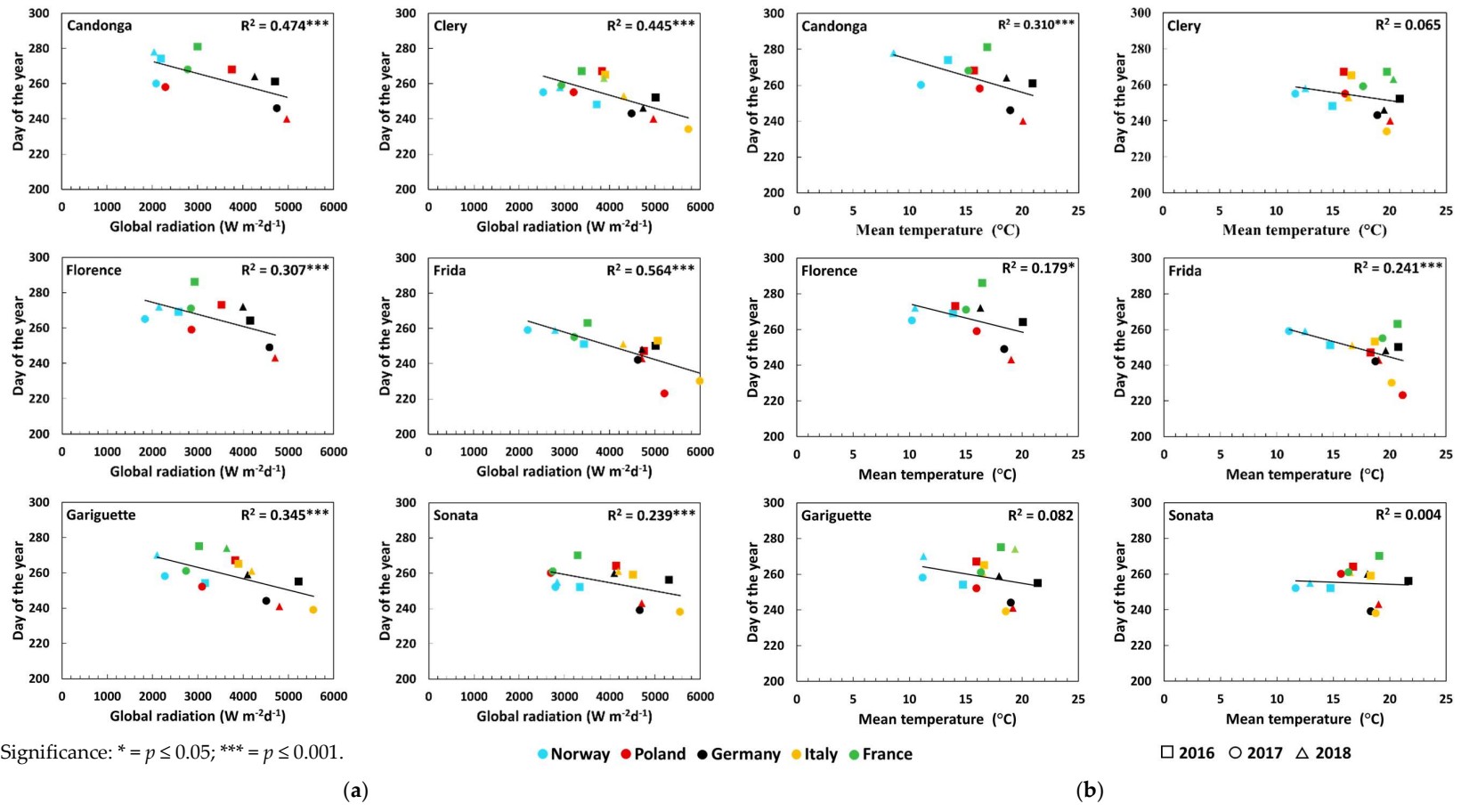

**Figure 3.** Day of flower initiation (floral developmental Stage 2) for six cultivars as influenced by (**a**) global radiation and (**b**) mean temperature during the 3-week period preceding flower initiation in 3 years.

To investigate whether the type of cultivation was affecting plant physiological responses, a regression analysis was performed (Table 2). In the studied cultivars, the time of FI was more enhanced by global radiation when grown in soil-less cultivation, while temperature had a stronger effect on the time of FI when plants were grown under open-field conditions. An exception was 'Sonata', where FI was significantly affected only by global radiation under soil-less conditions.

**Table 2.** Regression coefficients ($R^2$) for the date of flower initiation related to global radiation and temperature in six strawberry cultivars during the 3-week period preceding flower initiation when grown at five locations in Europe.

| | | Regression Coefficients ($R^2$) | | | | | |
|---|---|---|---|---|---|---|---|
| | | 'Candonga' | 'Clery' | 'Florence' | 'Frida' | 'Gariguette' | 'Sonata' |
| Global radiation ($Wm^{-2}d^{-1}$) | Open-field cultivation | 0.50 *** ↓ | 0.33 ** ↓ | 0.33 ** ↓ | 0.56 *** ↓ | 0.32 ** ↓ | 0.09 |
| | Soil-less cultivation | - | 0.75 *** ↓ | - | 0.72 *** ↓ | 0.62 *** ↓ | 0.66 *** ↓ |
| Temperature (°C) | Open-field cultivation | 0.46 *** ↓ | 0.30 ** ↓ | 0.36 *** ↓ | 0.56 *** ↓ | 0.37 *** ↓ | 0.02 |
| | Soil-less cultivation | - | 0.02 | - | 0.00 | 0.00 | 0.04 |

** = $p \le 0.01$; *** = $p \le 0.001$. The arrows indicate the direction of the relationship between flower initiation and global radiation and temperature: ↑ = positive, ↓ = negative. In Norway, Poland, and Germany, plants were grown under open-field cultivation; all cultivars were grown for 3 years. In Italy and France, plants were grown under soil-less cultivation; plants were propagated in trays. In Italy, all cultivars except 'Candonga' and 'Florence' were grown for 3 years, while in France, 'Candonga', 'Florence', 'Frida', and 'Sonata' were grown for 2 years, while 'Clery' and 'Gariguette' were grown for 3 year.

*3.2. Number of Crowns and Flowers*

Table 3 demonstrates the differences between cultivars and years for the number of crowns and flowers per plant at all experimental locations across Europe. In general, 'Florence' produced the highest number of flowers and crowns, while 'Candonga' had the lowest variability across all studied locations and years. For open-field cultivation, the highest mean number of crowns and flowers per plant for all cultivars and years were registered in Germany, while the plants of all cultivars grown in Norway and Poland had a lower average number of flowers and crowns. Since the late-flower-initiating cultivars 'Candonga' and 'Florence' were not grown in Italy, and crowns and flowers were not counted in the spring of the third year (2019) in France, it is difficult to draw definite conclusions for these locations.

To test the hypothesis that climatic conditions are affecting the number of crowns and flowers in strawberry plants, temperature ($T_{mean}$) and global radiation were related to the phenological observations across the experimental locations. The effect of the environment before and after FI was assessed as a factor. The observed relation between FI and climatic conditions within the locations was negligible as all cultivars experienced relatively similar conditions in each geographical site (Table 4). In France, the number of flowers and crowns was evaluated in 2 years only, and showed a strong year effect (Table 3). Therefore, the observed significant data for France are not comparable to the other locations.

However, when the physiological response of individual cultivars across the locations was studied (Figures 4 and 5), a stronger relationship was observed due to the extended variability of the climatic conditions across the continent. The number of flowers (Figure 4) was similarly affected by both temperature and global radiation 3 weeks before FI (correlation coefficient between the two parameters: r = 0.819, p = 0.001). For all cultivars except 'Frida', the number of flowers increased in locations with higher temperature and global radiation. The closest relationships were observed for 'Florence', 'Clery', and 'Gariguette'. Most cultivars were affected more strongly by temperature than by global radiation (Figure 4b). An exception was 'Candonga', which was more affected by global radiation than by temperature (Figure 4). Both global radiation and temperature affected

the number of flowers in all cultivars stronger under open-field growing conditions than under soil-less cultivation conditions (Table 5).

In contrast, climatic conditions 3 weeks before FI had a negligible effect on the number of crowns (Figure 5). However, regression analysis considering the type of cultivation conditions revealed interactions between cultivar, cultivation type, and climatic parameters (Table 5). In 'Clery', the number of crowns was positively affected by both increasing global radiation and temperature when grown under an open-field condition, while in 'Frida' and 'Sonata' under soil-less conditions, global radiation affected the number of crowns negatively. Moreover, at such conditions, the number of crowns in 'Frida' was positively influenced by temperature.

**Table 3.** Number of crowns and flowers per plant emerging in spring in six strawberry cultivars grown at five locations in Europe.

| Location | Year | Number of Crowns per Plant | | | | | | | Number of Flowers per Plant | | | | | | |
|---|---|---|---|---|---|---|---|---|---|---|---|---|---|---|---|
| | | Cand. | Cler. | Flor. | Frid. | Gari. | Sona. | | Cand. | Cler. | Flor. | Frid. | Gari. | Sona. | |
| Norway | 2017 | 4.0 | 3.8 | 6.3 | 3.9 | 3.5 | 3.9 | | 14.9 | 16.2 | 27.3 | 24.4 | 13.0 | 24.7 | |
| | 2018 | 3.6 | 2.6 | 4.7 | 4.2 | 3.3 | 3.9 | | 11.7 | 7.5 | 20.9 | 15.3 | 10.1 | 20.5 | |
| | 2019 | 7.1 | 3.7 | 9.0 | 10.0 | 5.5 | 6.6 | | 27.9 | 10.4 | 33.1 | 45.3 | 18.8 | 40.1 | |
| | *Mean* | *4.9* | *3.4* | *6.7* | *6.0* | *4.1* | *1.8* | *5.0 CD* | *18.2* | *11.4* | *27.1* | *28.4* | *14.2* | *28.4* | *21.3 A* |
| Poland | 2017 | 6.8 | 4.8 | 6.9 | 6.9 | 4.9 | 3.7 | | 13.5 | 10.5 | 36.1 | 36.5 | 19.9 | 17.9 | |
| | 2018 | 3.2 | 2.4 | 4.3 | 3.1 | 3.1 | 2.8 | | 13.8 | 11.7 | 33.9 | 23.5 | 17.2 | 21.1 | |
| | 2019 | 4.7 | 3.3 | 6.3 | 4.6 | 3.5 | 2.9 | | 19.4 | 21.0 | 45.1 | 46.1 | 30.1 | 28.6 | |
| | *Mean* | *4.9* | *3.5* | *5.8* | *4.9* | *3.8* | *3.1* | *4.4 B* | *15.6* | *14.4* | *38.8* | *35.4* | *22.4* | *22.3* | *24.8 AB* |
| Germany | 2017 | 7.1 | 7.9 | 10.3 | 10.3 | 6.5 | 6.7 | | 25.4 | 61.6 | 70.7 | 94.9 | 48.1 | 75.6 | |
| | 2018 | 6.5 | 6.3 | 9.8 | 5.6 | 5.0 | 4.5 | | 41.5 | 80.2 | 133.6 | 96.3 | 58.1 | 85.9 | |
| | 2019 | 6.1 | 6.0 | 7.5 | 6.7 | 6.7 | 5.1 | | 39.6 | 65.8 | 89.1 | 81.1 | 75.9 | 72.8 | |
| | *Mean* | *6.6* | *6.7* | *9.2* | *7.5* | *5.7* | *5.4* | *6.9 D* | *35.5* | *69.2* | *97.8* | *90.8* | *60.7* | *78.1* | *72.0 D* |
| Italy | 2017 | - | 2.5 | - | 2.4 | 3.7 | 2.1 | | - | 15.8 | - | 25.5 | 30.4 | 26.0 | |
| | 2018 | - | 2.5 | - | 1.7 | 3.5 | 2.4 | | - | 21.1 | - | 24.7 | 34.7 | 37.1 | |
| | 2019 | - | 3.4 | - | 2.1 | 4.2 | 2.1 | | - | 32.3 | - | 31.9 | 43.3 | 27.3 | |
| | *Mean* | *-* | *2.8* | *-* | *2.1* | *3.8* | *2.2* | *2.7 A* | *-* | *23.0* | *-* | *27.3* | *36.1* | *30.1* | *29.2 B* |
| France | 2017 | 7.5 | 4.1 | 8.8 | 6.3 | 3.5 | 6.2 | | 31.9 | 31.5 | 71.3 | 48.5 | 40.1 | 58.5 | |
| | 2018 | 2.9 | 2.9 | 6.1 | 3.5 | 2.2 | 4.2 | | 11.2 | 20.3 | 43.4 | 11.0 | 19.9 | 39.7 | |
| | 2019 | - | - | - | - | - | - | | - | - | - | - | - | - | |
| | *Mean* | *5.2* | *3.5* | *7.4* | *4.9* | *2.8* | *5.2* | *4.9 C* | *21.6* | *25.9* | *57.3* | *29.8* | *30.3* | *49.1* | *35.7 C* |
| *Cultivar mean* | | *5.4 b* | *4.0 a* | *7.3 c* | *5.1 b* | *4.2 a* | *4.1 a* | | *22.8 a* | *28.9 b* | *60.0 d* | *43.2 c* | *32.9 b* | *41.1 c* | |
| Probability level of significance (ANOVA) | | | | | | | | | | | | | | | |
| Source of variation | | | | | | | | | | | | | | | |
| Location (A) | | <0.001 | | | | | | | <0.001 | | | | | | |
| Cultivar (B) | | <0.001 | | | | | | | <0.001 | | | | | | |
| Year © | | <0.001 | | | | | | | <0.001 | | | | | | |
| A × B | | <0.001 | | | | | | | <0.001 | | | | | | |
| A × C | | <0.001 | | | | | | | <0.001 | | | | | | |
| B × C | | 0.088 | | | | | | | 0.007 | | | | | | |
| A × B × C | | <0.001 | | | | | | | <0.001 | | | | | | |

Values for cultivar mean followed by different lowercase letters indicate a significant difference between cultivars by Tukey, $p \leq 0.05$. Values for location mean followed by different capital letters indicate a significant difference between cultivars by Tukey, $p \leq 0.05$. Each individual value comprises 15 measurements. Cultivars: Cand. = 'Candonga'; Cler. = 'Clery'; Flor. = 'Florence'; Frid = 'Frida'; Gari. = 'Gariguette'; Sona. = 'Sonata'.

**Table 4.** Regression coefficient ($R^2$) for the number of flowers and crowns as an average of six strawberry cultivars related to global radiation and temperature during the 3-week period preceding and 5 weeks after flower initiation (FI) when grown at five locations in Europe.

| | | | Regression coefficient ($R^2$) | | | | |
|---|---|---|---|---|---|---|---|
| | | | **Norway** | **Poland** | **Germany** | **Italy** | **France** |
| Three weeks before FI | Number of flowers | Global radiation ($Wm^{-2}d^{-1}$) | 0.01 | 0.11 * ↑ | 0.01 | 0.00 | 0.05 |
| | | Temperature (°C) | 0.00 | 0.07 * ↑ | 0.10 * ↓ | 0.05 ↓ | 0.01 ↑ |
| | Number of crowns | Global radiation ($Wm^{-2}d^{-1}$) | 0.05 | 0.03 | 0.01 | 0.08 | 0.06 |
| | | Temperature (°C) | 0.06 | 0.03 | 0.06 | 0.15 * ↓ | 0.00 |
| Five weeks after FI | Number of flowers | Global radiation ($Wm^{-2}d^{-1}$) | 0.01 | 0.15 ** ↑ | 0.03 | 0.02 | 0.32 *** ↓ |
| | | Temperature (°C) | 0.01 | 0.10 * ↑ | 0.03 | 0.04 | 0.34 *** ↓ |
| | Number of crowns | Global radiation ($Wm^{-2}d^{-1}$) | 0.03 | 0.03 | 0.15 ** ↓ | 0.06 | 0.49 *** ↓ |
| | | Temperature (°C) | 0.15 * ↓ | 0.10 * ↓ | 0.02 | 0.05 | 0.53 *** ↓ |

* = $p \leq 0.05$; ** = $p \leq 0.01$; *** = $p \leq 0.001$. The arrows in the columns indicate the direction of the relationship between flower initiation and global radiation and temperature: ↑ = positive, ↓ = negative. Open-field cultivation conducted in Norway, Poland, and Germany; all cultivars were evaluated for 3 years. In Norway, Poland, and Germany, plants were grown under open-field cultivation; all cultivars were grown for 3 years. Soil-less cultivation conducted in Italy and France, and plants were raised in trays. In Italy, all cultivars except 'Candonga' and 'Florence' were grown and evaluated for 3 years, while in France, all cultivars were evaluated for 2 years (2017 and 2018).

**Table 5.** Regression coefficients ($R^2$) for six strawberry cultivars for their number of flowers and crowns related to global radiation and temperature during the 3-week period preceding flower initiation when grown at five locations in Europe.

| | | | Regression Coefficients | | | | | |
|---|---|---|---|---|---|---|---|---|
| | | | **Candonga** | **Clery** | **Florence** | **Frida** | **Gariguette** | **Sonata** |
| Number of flowers | Global radiation ($Wm^{-2}d^{-1}$) | Open-field cultivation | 0.23 * ↑ | 0.50 *** ↑ | 0.48 *** ↑ | 0.21 * ↑ | 0.36 *** ↑ | 0.32 ** ↑ |
| | | Soil-less cultivation | - | 0.01 | - | 0.01 | 0.10 | 0.17 |
| | Temperature (°C) | Open-field cultivation | 0.12 | 0.52 *** ↑ | 0.35 *** ↑ | 0.24 * ↑ | 0.37 *** ↑ | 0.34 ** ↑ |
| | | Soil-less cultivation | - | 0.02 | - | 0.02 | 0.12 | 0.16 |
| Number of crowns | Global radiation ($Wm^{-2}d^{-1}$) | Open-field cultivation | 0.14 | 0.48 *** ↑ | 0.15* ↑ | 0.00 | 0.11 | 0.04 |
| | | Soil-less cultivation | - | 0.15 | - | 0.46 ** ↓ | 0.21 | 0.43 ** ↓ |
| | Temperature (°C) | Open-field cultivation | 0.03 | 0.44 *** ↑ | 0.11 | 0.00 | 0.09 | 0.02 |
| | | Soil-less cultivation | - | 0.04 | - | 0.30 * ↑ | 0.02 | 0.06 |

* = $p \leq 0.05$; ** = $p \leq 0.01$; *** = $p \leq 0.001$. The arrows in the columns indicate the direction of the relationship between flower initiation and global radiation and temperature: ↑ = positive, ↓ = negative. Open-field cultivation conducted in Norway, Poland, and Germany; all cultivars were evaluated for 3 years. In Norway, Poland, and Germany, plants were grown under open-field cultivation; all cultivars were grown for 3 years. Soil-less cultivation conducted in Italy and France, and plants were raised in trays.

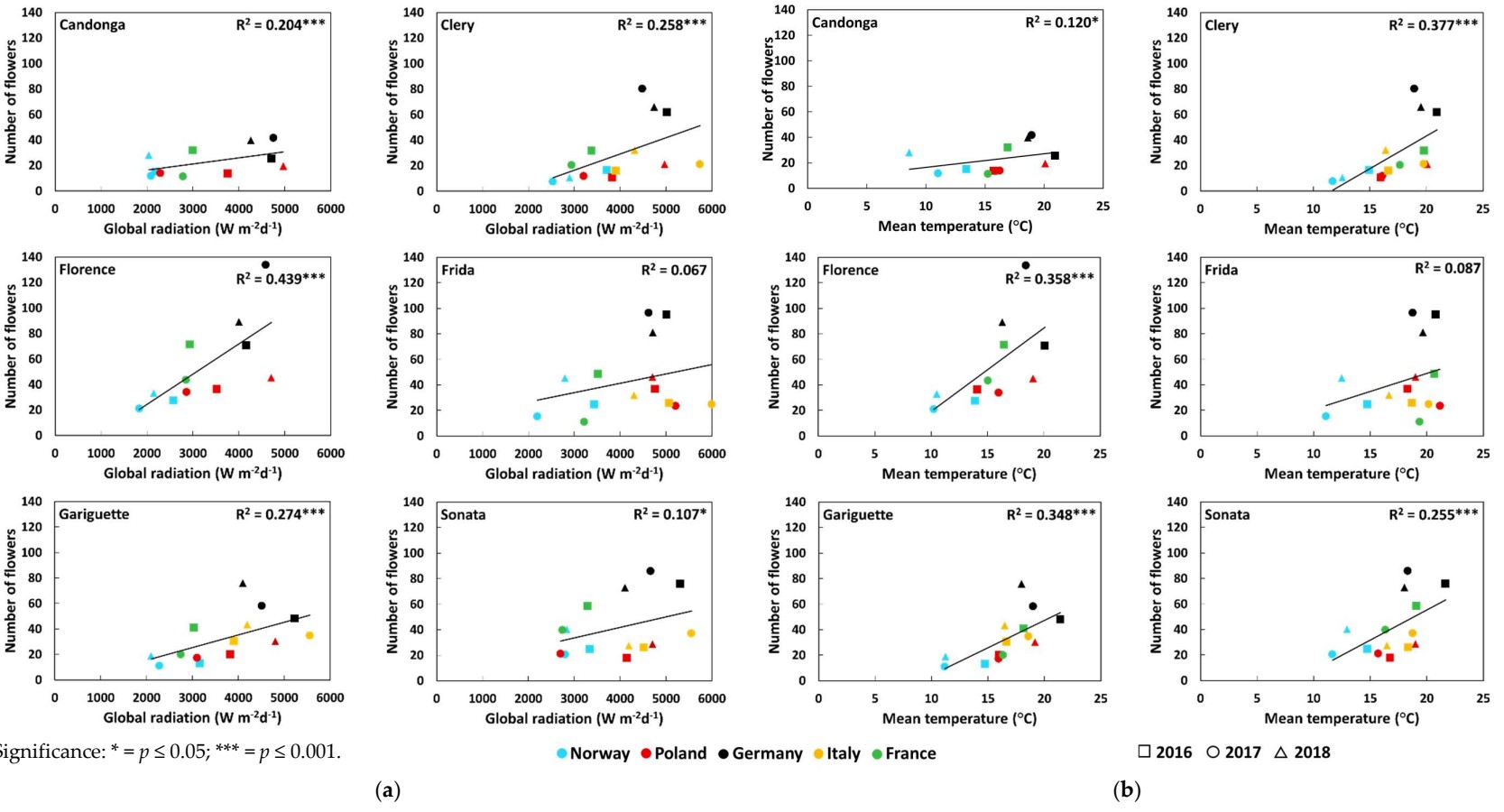

**Figure 4.** Number of flowers of six cultivars grown at five locations in Europe as influenced by (**a**) global radiation and (**b**) mean temperature during the 3-week period preceding flower initiation in 3 years.

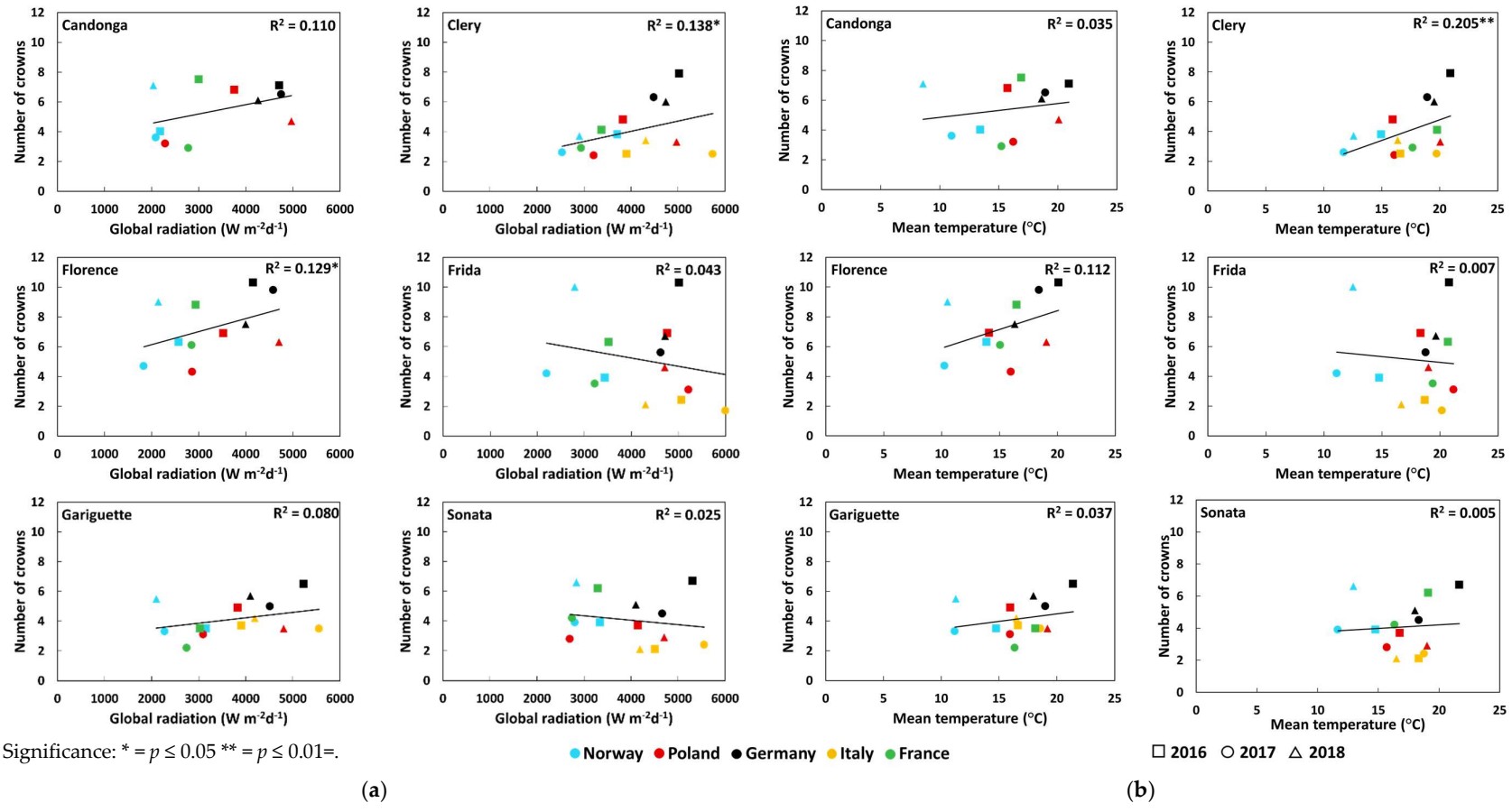

**Figure 5.** Number of crowns of six cultivars grown at five locations in Europe as influenced by (**a**) global radiation and (**b**) mean temperature during the 3-week period preceding flower initiation in three years.

In Italy, all cultivars except 'Candonga' and 'Florence' were grown and evaluated for 3 years, while in France, all cultivars were evaluated for 2 years (2017 and 2018). As a consequence, no regression analyses were performed for 'Candonga' and 'Florence under soil-less cultivation. The number of flowers and crowns in all cultivars (Figures 6 and 7) was also affected by both temperature and global radiation during the 5 weeks after FI (correlation coefficient between the two factors in this period: r = 0.839, *p* = 0.001). Both climatic factors during this period had a stronger effect on the number of flowers than on crowns. A positive relationship between the number of flowers and global radiation was observed for all cultivars except 'Frida' (Figure 6a) with the strongest effect on 'Gariguette', followed by 'Florence' and 'Clery'. For all cultivars except 'Candonga' and 'Frida', the number of flowers was also positively affected by temperature with 'Clery' showing the strongest relationship (Figure 6b).

The effects of global radiation and temperature on the number of crowns were negligible for all cultivars (Figure 7). However, the regression analysis of the cultivation conditions (Table 6) revealed interactions between cultivar, cultivation type, and climatic parameters. Under open-field conditions, the number of flowers was positively related to global radiation and temperature in all cultivars, with 'Candonga' and 'Frida' only showing weak relationships. Under soil-less conditions, the number of flowers was only affected in 'Sonata'.

**Table 6.** Regression coefficients ($R^2$) for six strawberry cultivars for their number of flowers and crowns related to global radiation and temperature during the 5-week period preceding flower initiation when grown at five locations in Europe.

| | | | Regression coefficients ($R^2$) | | | | | |
| --- | --- | --- | --- | --- | --- | --- | --- | --- |
| | | | **Candonga** | **Clery** | **Florence** | **Frida** | **Gariguette** | **Sonata** |
| Number of flowers | Global radiation ($Wm^{-2}d^{-1}$) | Open-field cultivation | 0.26 * ↑ | 0.49 *** ↑ | 0.36 *** ↑ | 0.23 * ↑ | 0.50 *** ↑ | 0.41 *** ↑ |
| | | Soil-less cultivation | - | 0.15 | - | 0.03 | 0.18 | 0.33* ↓ |
| | Temperature (°C) | Open-field cultivation | 0.18 * ↑ | 0.54 *** ↑ | 0.37 *** ↑ | 0.20 * ↑ | 0.42 *** ↑ | 0.45 *** ↑ |
| | | Soil-less cultivation | - | 0.01 | - | 0.01 | 0.00 | 0.29 * ↑ |
| Number of crowns | Global radiation ($Wm^{-2}d^{-1}$) | Open-field cultivation | 0.01 | 0.23 * ↑ | 0.04 | 0.00 | 0.02 | 0.02 |
| | | Soil-less cultivation | - | 0.09 | - | 0.68 *** ↓ | 0.04 | 0.60 *** ↓ |
| | Temperature (°C) | Open-field cultivation | 0.01 | 0.28 ** ↑ | 0.02 | 0.01 | 0.01 | 0.00 |
| | | Soil-less cultivation | - | 0.10 | - | 0.08 | 0.40 * ↓ | 0.34 * ↑ |

\* = *p* ≤ 0.05; \*\* = *p* ≤ 0.01; \*\*\* = *p* ≤ 0.001. The arrows in the columns indicate the direction of the relationship between flower initiation and global radiation and temperature: ↑ = positive, ↓ = negative. Open-field cultivation conducted in Norway, Poland, and Germany; all cultivars were evaluated for 3 years. In Norway, Poland, and Germany, plants were grown under open-field cultivation; all cultivars were grown for 3 years. Soil-less cultivation conducted in Italy and France, and plants were raised in trays. In Italy, all cultivars except 'Candonga' and 'Florence' were grown and evaluated for 3 years, while in France, all cultivars were evaluated for 2 years (2017 and 2018). As a consequence, no regression analyses were performed for 'Candonga' and 'Florence under soil-less cultivation.

In 'Clery', the number of crowns (Table 6) was positively influenced by global radiation and temperature under open-field conditions. 'Frida' and 'Sonata' were negatively affected by global radiation after FI only in soil-less production. Under this growing condition, the effect of temperature was not consistent: while the number of crowns decreased in 'Gariguette' with increasing temperature, it increased in 'Sonata', which is in line with the influence of global radiation.

Finally, an assessment of the relationships between GDD with 5 °C as base temperature and the number of flowers and crowns was performed for the 3-week period preceding anthesis in spring (Figure 8). Higher GDD during this period was associated with reduced numbers of flowers in 'Frida' and 'Clery' (Figure 8a), as well as crowns in all cultivars except 'Florence' (Figure 8b). This phenomenon is probably linked to the large climatic differences prevailing from the restart of growth in spring until anthesis at the various locations across the European continent. For example, it can be seen from Figure A3 that for the Norwegian and Polish locations, the date of anthesis is associated with highly variable weather conditions between the years.

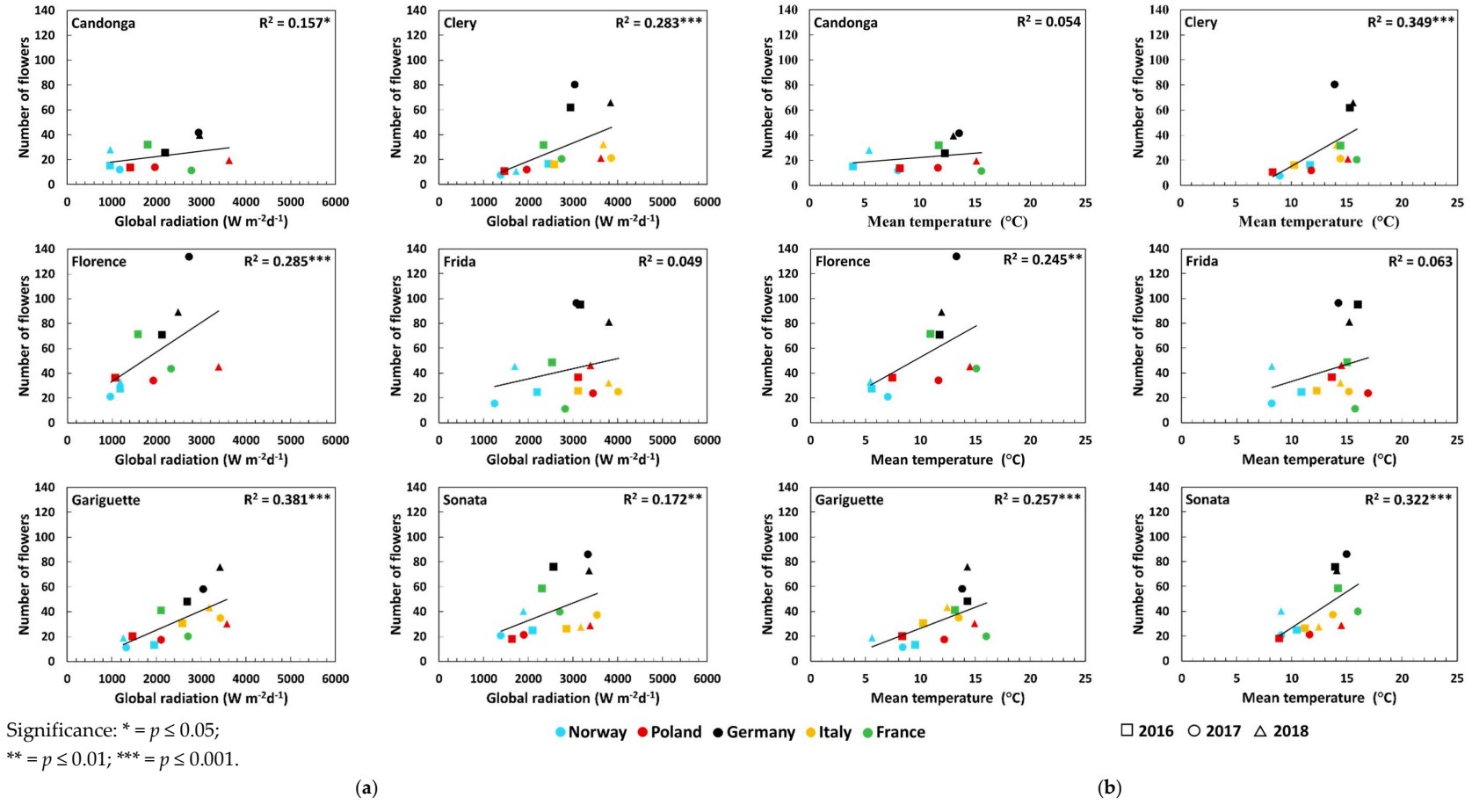

Significance: * = *p* ≤ 0.05;
** = *p* ≤ 0.01; *** = *p* ≤ 0.001.

● Norway ● Poland ● Germany ● Italy ● France

□ 2016 ○ 2017 △ 2018

(**a**)

(**b**)

**Figure 6.** Number of flowers of six cultivars grown at five locations in Europe as influenced by (**a**) global radiation and (**b**) mean temperature during the 5-week period after flower initiation in 3 years.

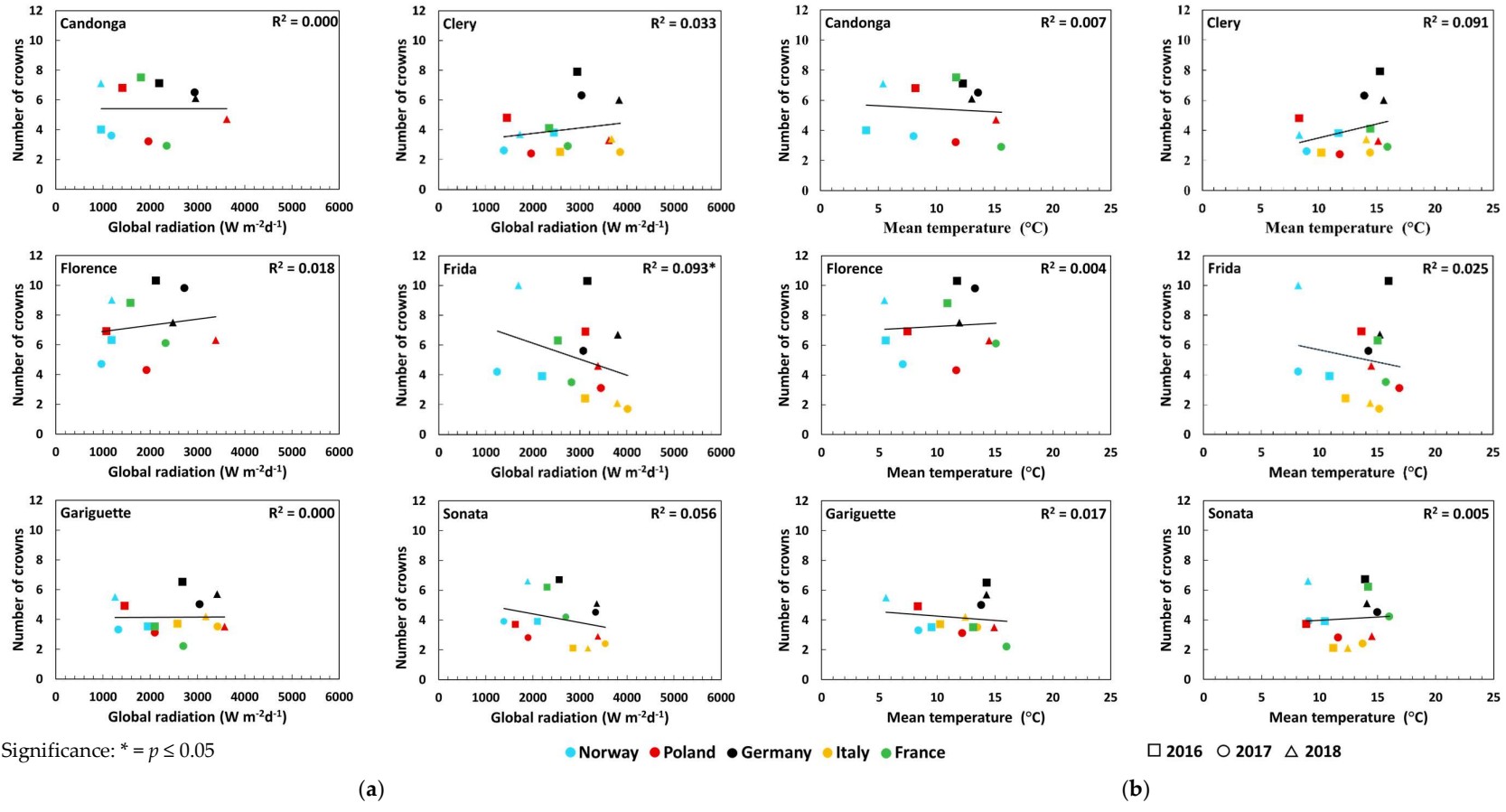

**Figure 7.** Number of crowns of six cultivars grown at five locations in Europe as influenced by (**a**) global radiation and (**b**) mean temperature during the 5-week period after flower initiation in 3 years.

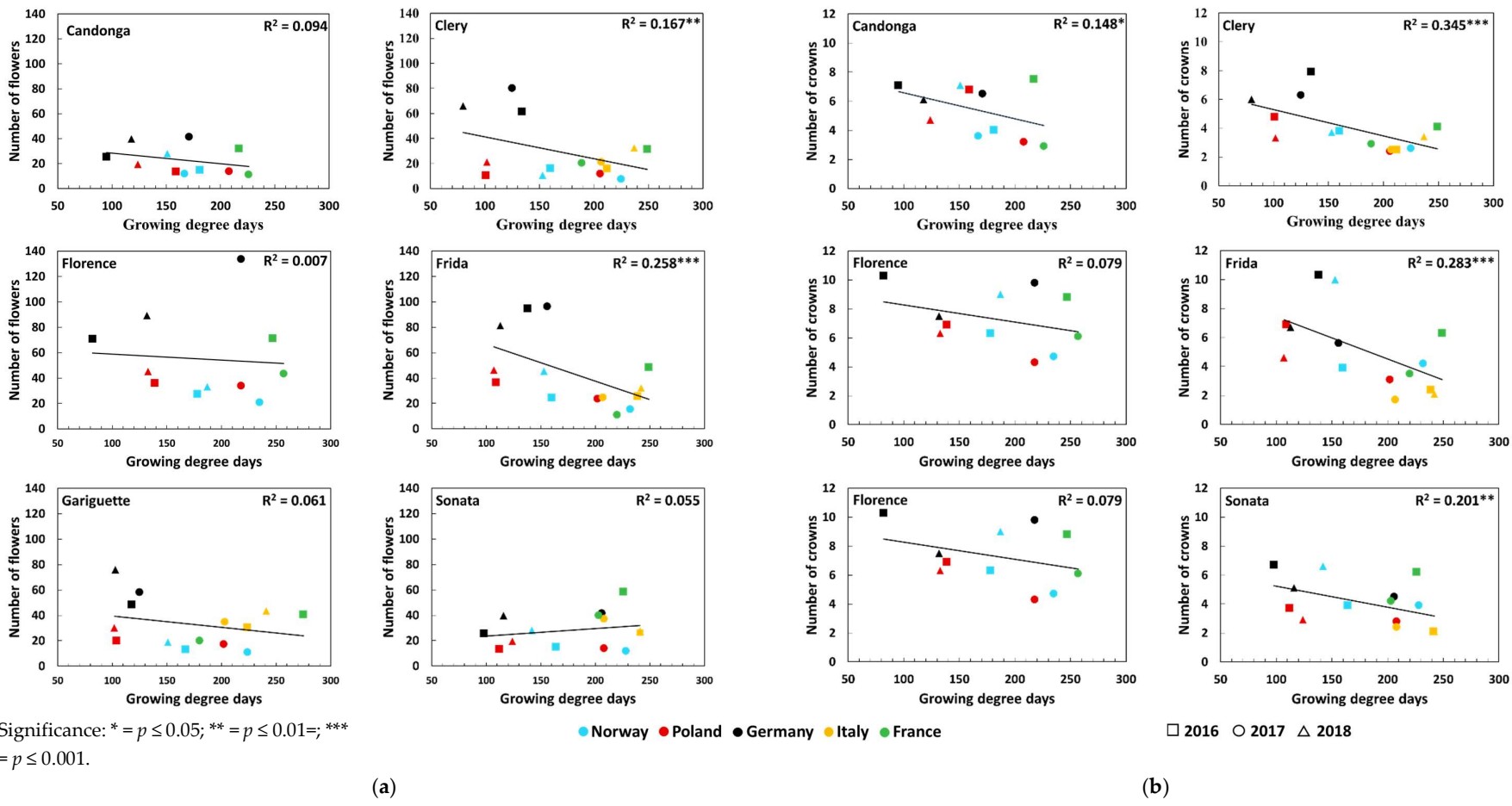

Significance: * = *p* ≤ 0.05; ** = *p* ≤ 0.01=; *** = *p* ≤ 0.001.

● Norway ● Poland ● Germany ● Italy ● France          □ 2016  ○ 2017  △ 2018

(**a**)                                                                                                    (**b**)

**Figure 8.** Number of flowers (**a**) and crowns (**b**) of six strawberry cultivars grown at five locations in Europe as influenced by the growing degree-day accumulation during the 3-week period preceding anthesis in spring in 3 years.

## 4. Discussion

In the present study, we investigated the environmental effects on the phenological patterns of six genetically distant strawberry cultivars across a range of geographic locations in Europe. The wide latitudinal range used (45 to 60° N) provided unique possibilities to study cultivar responses to the prevailing local climatic conditions. The obtained results are generally consistent with previous findings regarding the effect of temperature and radiation on the time of flower initiation (FI) and the number of flowers and crowns obtained in strawberry under controlled conditions [2,3,5,6,10,12,14].

Photoperiod is the primary environmental factor controlling the transition from vegetative to reproductive growth in strawberry, and cultivars are commonly classified based on their photoperiodic sensitivity [3]. However, due to the pronounced photoperiod x temperature interaction in the process, temperature can be as important as photoperiod for floral initiation, especially at high latitudes with long photoperiods [5]. In general, for all cultivars and years, DOY-FI was delayed in Norway and France in comparison with the other locations (Figure 1). In Norway, this was due to a combination of low temperature and photoperiod, which was above the critical length (≤14 h) until the beginning of September, while in France, the delayed FI was caused by the high temperature in August and until mid-September.

As previously shown, FI takes place over a limited range of temperatures, which also determines the number of flowers per plant and per inflorescence under flower-inducing photoperiodic conditions [2,3]. Temperature and global radiation are always highly intercorrelated in nature, and therefore, their effects are difficult to separate under natural field conditions. Moreover, the determination of the effect of the photoperiod on DOY-FI is not possible because of the inherent relationship between DOY and photoperiod (decreasing or increasing gradually day by day). However, experiments under controlled conditions have shown that a high total daily light integral can significantly enhance FI [2]. The present study confirms that increasing global radiation in the 3-week period preceding FI enhanced floral initiation (Figure 2). Takeda et al. [15] showed that the use of photoselective shading nets postponed the flowering of strawberry. However, the authors did not state whether the delay in flowering time was related to the restriction of the FI process or organogenesis. In the present study, FI was later in locations with lower global radiation at comparable temperatures (Figure 1, Germany versus France). When compared with Geisenheim, a higher summer temperature was observed in France, and that was apparently the main reason for the postponed DOY-FI. The local French observation indicated that the initiation of flowers in strawberry is delayed by higher summer temperatures, which confirms the known threshold role of temperature in triggering the flowering initiation. These two factors, temperature and global radiation, combined, are influencing DOY-FI, but their impacts are hardly separable under natural conditions.

It was also found that in three everbearing strawberry cultivars, there was a positive correlation between the photosynthetic rate and flower numbers [14]. This agrees with previous studies, where light intensity positively affected flower development in strawberry [12]. Yoshida et al. [16] also found that intermittent cold storage in the dark advanced floral initiation in autumn due to increased photosynthesis and carbohydrate content of the plants. It is thus evident that adequate photosynthetic conditions and nonlimiting plant carbohydrate content are basically important for FI and the development in seasonal flowering strawberries. Furthermore, there is some evidence indicating that high sucrose content in the crowns may affect FI directly by acting as a physiological signal for FI in strawberry and several other photoperiodic plants [17–19].

The present results clearly demonstrate that temperature and global radiation, which are intercorrelated environmental variables, are important modifying factors for the determination of the number of flowers and, to a lesser extent, crowns in SF (Figures 4–7). The effects were stronger and more pronounced under open-field conditions than under soil-less conditions (Tables 5 and 6). However, it should be noted that the use of both

cultivation types also represents different locations from north to south of Europe and thus may be more an effect of the environmental conditions than of the type of cultivation itself.

The results also demonstrate that with the present planting date (week no. 32) for open-field production, the combination of temperature and global radiation has been optimal at Geisenheim, Germany, and least favorable in Norway. However, if an earlier planting date had been chosen, the results would probably have been quite different. Thus, it is a longtime experience in Norway that planting later than 15 July will not give an acceptable berry yield in the following cropping season because of inadequate time for the establishment of a plant with enough potential flowering sites. To some extent, the same applies to the situation in Poland. This means that the present results do not indicate the general yield potential of the various experimental locations, since this would have required adjustment of the planting date to the local conditions, an alternative that would not have been practically possible to establish in order to standardize the trial conditions. Moreover, it should be noticed that, in Italy, the two late-initiating cultivars, 'Candonga' and 'Florence', were not included and thus did not contribute to the location mean. In addition, in both Italy and France, plants were grown in trays in the field (until cold-stored in December). Growth conditions in trays with their limited root volume may thus have affected the FI of the plants [2,3], as well as their general response to environmental conditions (differences presented in Tables 5 and 6).

In addition to affecting FI in strawberry [2,3,20], temperature is a fundamental factor for controlling flower development and the rate of fruit ripening. Thus, it is a common experience from both commercial practice and research that the earliness of flowering and fruit ripening is mainly determined by the accumulated heat sum (GDD) in spring and early summer. In the present study, however, there was also a consistent negative effect of high spring temperature on the number of crowns and flowers that reached anthesis (Figure 8). This is an effect that might be due to an advantage of the most advanced floral primordia and partial abortion among the later ones. In other words, high spring temperature might initiate growth and development of the most advanced primordia at the expense of those lagging behind [21].

The importance of September temperature for fruit yield in the subsequent season was also demonstrated by Døving and Måge [22]. Their analysis of data for the period 1975–2000 for the district of Valldal in mid-Norway revealed a significant ($p = 0.005$) positive correlation between September temperature and subsequent year fruit yields of 'Senga Sengana' strawberry. The effect was markedly enhanced when combined with low August temperature. This effect was obviously an expression of the suboptimal September temperature for SD floral initiation in strawberry in high-latitude regions [3]. In addition, for plums, an overall positive correlation was observed between August, September, and August plus September temperature and the amount of flowering in the subsequent spring [23].

However, the high number of crowns and flowers in Germany may not only be due to the optimal combination of temperature and global radiation for FI and development but may also be influenced by their common practice of irrigation from mid-September to mid-October and in early spring, already before anthesis [24].

In the present study, in addition to the effect of $T_{mean}$, we analyzed also $T_{min}$ and $T_{max}$ for the various developmental processes in strawberry, but for all major processes studied, $T_{mean}$ gave as good a relationship as did $T_{min}$ and $T_{max}$ (data not shown). While in controlled environments, the temperature can be widely manipulated in such a way that the dark (night) temperature may exceed the critical temperatures for FI; it seems that under field conditions, the diurnal temperature fluctuations are seldom large enough to exceed these limits. Therefore, under field conditions, there is no need for such elaborate analysis of the various temperature components.

In the present study, we also compared the physiological response among the various strawberry cultivars as characterized by their contrasting local background. For example, the analysis suggests that 'Sonata' from the Netherlands is a widely adapted cultivar with

low dependency on external environmental stimuli for the date of FI (Figure 3, Table A2). On the other hand, the different response of 'Frida' (originating from Norway) under Nordic conditions (i.e., 'Frida' was the earliest cultivar in all locations except in Norway) might be explained by the slower temperature accumulation in the north, combined with a longer critical photoperiod for 'Frida' compared with cultivars of lower-latitude origin [14]. This would give 'Frida' an FI advantage early in the season in the south when high temperature is not limiting, but a disadvantage at the higher temperatures prevailing there later in the season (temperature x daylength interaction effects). In addition, rapid changes in daylength in the north may play an additional role during the FI process. Therefore, it is possible that the 'temperature x daylength' interaction may force the physiological process of such a northern cultivar in a slightly different way in the south compared with that under northern conditions (Figure 1).

Moreover, the number of flowers and, less pronounced, crowns demonstrates cultivar-specific interactions with global radiation and temperature. In 'Clery', originating from Italy and with early FI, the number of flowers and crowns was strongly positively affected by both global radiation and temperature before and after FI in open-field locations (Table 5 and 6). Thus, 'Clery' showed only low plasticity for the environmental factors studied when related to the yield-determining number of flowers and crowns. On the one hand, only low effects were observed for 'Candonga' and 'Frida', two cultivars of contrasting origin (Spain and Norway), and for the date of FI (late and early). The observed results may indicate that these cultivars may be adapted to a wide range of growing conditions. In 'Florence', only the number of flowers was positively related to global radiation and temperature before and after FI, while the number of crowns was unaffected. A strong temperature effect on the yield of 'Florence' was also discussed in previous studies [14,20] for this cultivar, being responsible for the low yield in the northern part of Norway. However, no differentiation was made in these studies between the yield-influencing numbers of flowers and crowns per plant.

The present analysis of the internal relationships between climatic variables and phenological development also highlights the difficulties encountered for the interpretation of such data obtained across Europe. To better verify the above-mentioned relationships, more than 3 years of study is necessary. Such complex data require a sophisticated model to clearly identify cultivar x environmental x type of cultivation interactions, which will be of high interest in the context of global warming and changes in cultivation techniques already implemented or necessary in the future.

### 5. Conclusions

The results of the study demonstrate that combined analyses of phenological and climatic data obtained from the various locations across Europe can be very useful for deciphering the complex relationship between climate and dynamic developmental traits, such as flower initiation and generative development in strawberry. Besides the well-known temperature effect, intercorrelated global radiation may influence these processes and needs to be studied further. In the present study, early FI was associated with elevated temperature and global radiation. The cultivar 'Frida' revealed the highest dependency on global radiation for flower initiation, while 'Sonata' was least affected by temperature and radiation. In general, temperature and global radiation in periods both preceding and following flower initiation had a stronger positive effect on the number of flowers than on crowns, especially under open-field conditions. The influence of these factors was highly variable across the cultivars: 'Clery', 'Florence', and 'Gariguette were most affected, while 'Frida' was least influenced.

**Author Contributions:** Conceptualization, E.K., A.S.; methodology, E.K, A.S., B.D.; data curation, T.L.W., E.K., K.K.; funding acquisition, E.K., A.S., B.D., A.M., G.S.; investigation, A.S., R.R., A.M., I.S., E.K., B.B., K.E., D.M., G.S., B.D., M.D., K.G., A.P.; writing—original draft, O.M.H., T.L.W., A.S., E.K.; writing—review and editing, T.L.W., A.S., E.K., O.M.H., A.M., D.M., G.S., B.D., A.P. All authors have read and agreed to the published version of the manuscript.

**Funding:** This work was founded by the European Union's H2020 Programme (GoodBerry; grant number 679303). We acknowledge support from the Open Access Publishing Fund of Geisenheim University.

**Data Availability Statement:** Not applicable.

**Acknowledgments:** The authors would like to thank all farmworkers and laboratory staff for their skillful field management, sample collection, and analyses.

**Conflicts of Interest:** The authors declare no conflict of interest.

## Appendix A

**Table A1.** Mean temperature, global radiation, and precipitation at five locations in Europe for the period before and after flower initiation (1 June–30 November) and before anthesis (1 March–30 June).

|  |  | Year | Norway | Poland | Germany | Italy | France |
|---|---|---|---|---|---|---|---|
| Period before and after flower initiation 1 June–30 November | Mean temperature (°C) | 2016 | 9.4 | 12.7 | 15.0 | 13.5 | 17.1 |
|  |  | 2017 | 8.9 | 13.4 | 14.4 | 13.1 | 16.1 |
|  |  | 2018 | 10.8 | 14.0 | 16.1 | 14.3 | 17.7 |
|  | Global radiation (W m$^{-2}$d$^{-1}$) | 2016 | 2515 | 3031 | 3420 | 3734 | 3280 |
|  |  | 2016 | 2451 | 2853 | 3181 | 3901 | 2941 |
|  |  | 2018 | 2819 | 3404 | 3901 | 3850 | 3461 |
|  | Precipitation (mm) | 2016 | 303 | 320 | 171 | n.a. | 202 |
|  |  | 2017 | 400 | 401 | 286 | n.a. | 267 |
|  |  | 2018 | 237 | 360 | 100 | n.a. | 247 |
| Period before anthesis 1 March–30 June | Mean temperature (°C) | 2017 | 7.0 | 11.2 | 13,8 | 17.2 | 17.6 |
|  |  | 2018 | 7.3 | 12.1 | 14.0 | 16.4 | 16,2 |
|  |  | 2019 | 7.0 | 12.3 | 13.4 | 17.1 | * |
|  | Global radiation (W m$^{-2}$d$^{-1}$) | 2017⁻ | 3729 | 4124 | 4851 | 4924 | 4014 |
|  |  | 2019 | 4556 | 4778 | 4751 | 4399 | 3645 |
|  |  | 2019 | 3788 | 4739 | 4860 | 4450 | * |
|  | Precipitation (mm) | 2017 | 172 | 280 | 143 | n.a. | 289 |
|  |  | 2018 | 136 | 123 | 117 | n.a. | 394 |
|  |  | 2019 | 280 | 136 | 158 | n.a. | - |

n.a. = no data available; * no studies performed.

**Table A2.** Day of the year for strawberry floral development (Stage 2) as a mean of 3 years for six cultivars and five locations in Europe.

| Location | Candonga | Clery | Florence | Frida | Gariguette | Sonata | *Location Mean* |
|---|---|---|---|---|---|---|---|
| Norway | 271 | 254 | 269 | 256 | 261 | 253 | *260.5 B* |
| Poland | 255 | 254 | 258 | 238 | 253 | 256 | *252.4 A* |
| Germany | 257 | 247 | 262 | 248 | 253 | 252 | *252.8 A* |
| Italy | - | 251 | - | 245 | 255 | 253 | *250.8 A* |
| France | 275 | 263 | 278 | 259 | 270 | 265 | *268.2 C* |
| *Cultivar mean* | *263.5 cd* | *253.7 ab* | *265.7 d* | *248.2 a* | *258.4 bc* | *255.0 b* |  |
| Probability level of significance (ANOVA) |  |  |  |  |  |  |  |
| Source of variation |  |  |  |  |  |  |  |
| Cultivar (A) | ≤0.001 |  |  |  |  |  |  |
| Location (B) | ≤0.001 |  |  |  |  |  |  |
| A×B | ns |  |  |  |  |  |  |

Values are means of 3 years, each comprising 4 × 3 measurements. Values for cultivar mean followed by different lowercase letters indicate a significant difference between cultivars (Tukey-HSD, $p \leq 0.05$). Values for location mean followed by different capital letters indicate a significant difference between locations (Tukey-HSD, $p \leq 0.05$).

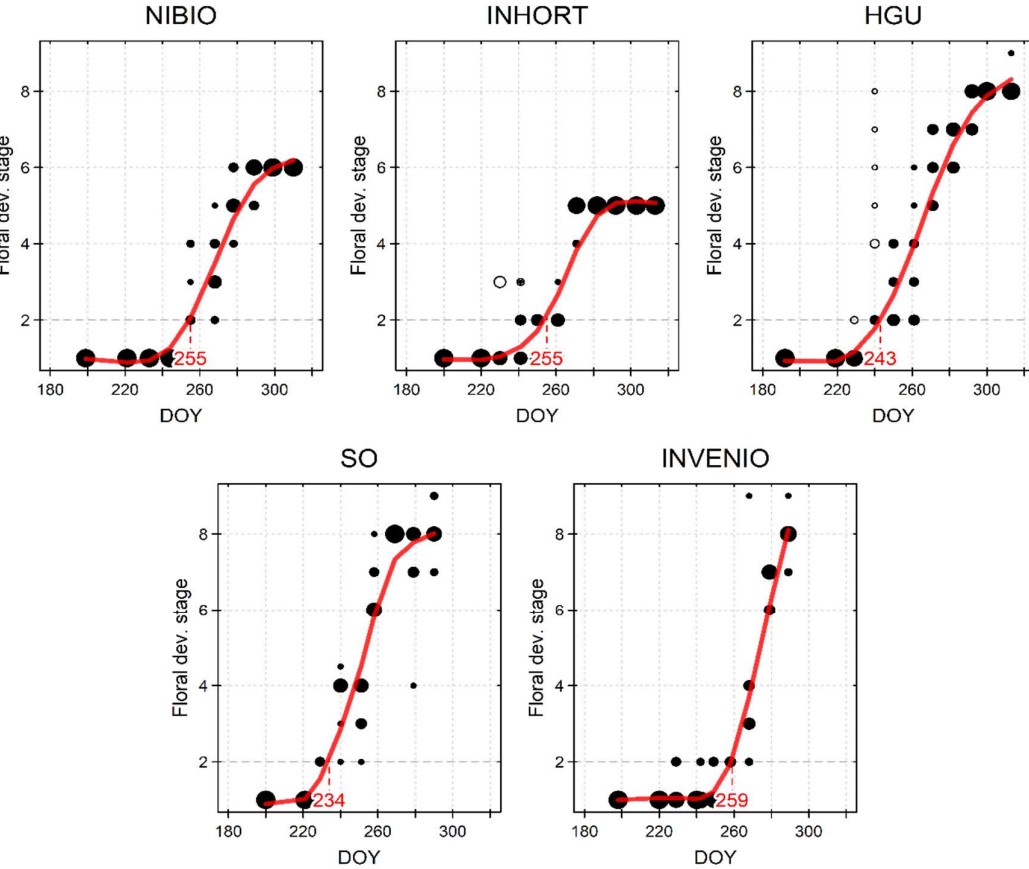

**Figure A1.** Exemplary calculation of the estimated day of the year (DOY, red number) for flower initiation at floral developmental Stage 2, performed by the cubic spline smoothing function (red line) over the registered floral developmental stages (black dots) at a given DOY for 'Clery' grown at five locations in Europe in 2017. The size of the dots represents the frequency of the registered values, and hollow dots represent outlying samples (influencing the shape of the function) that were not included in the calculation.

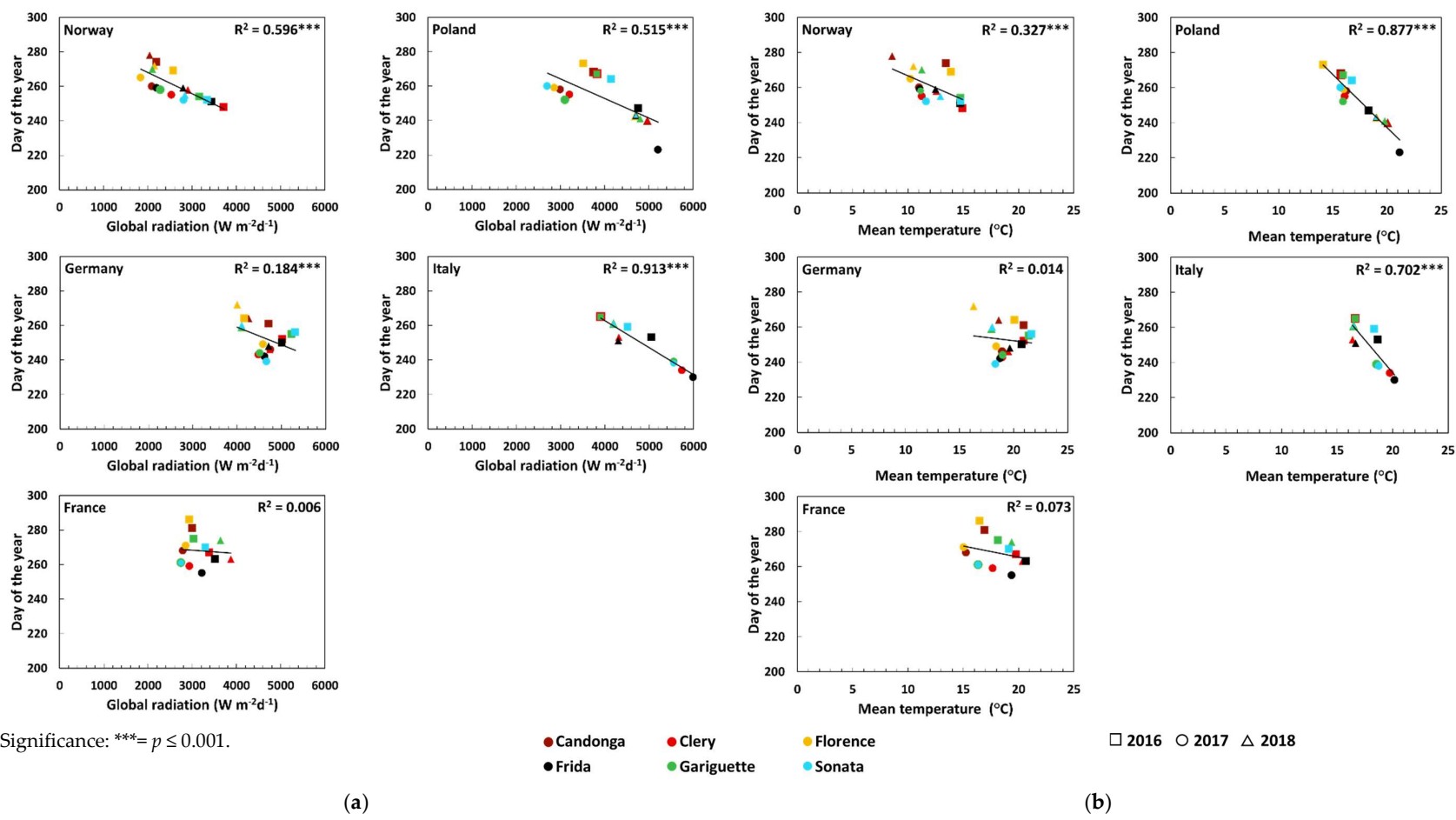

**Figure A2.** Day of the year for strawberry flower initiation (floral developmental Stage 2) for six strawberry cultivars grown at five locations in Europe as influenced by (**a**) global radiation and (**b**) mean temperature during the 3-week period preceding flower initiation for 3 years.

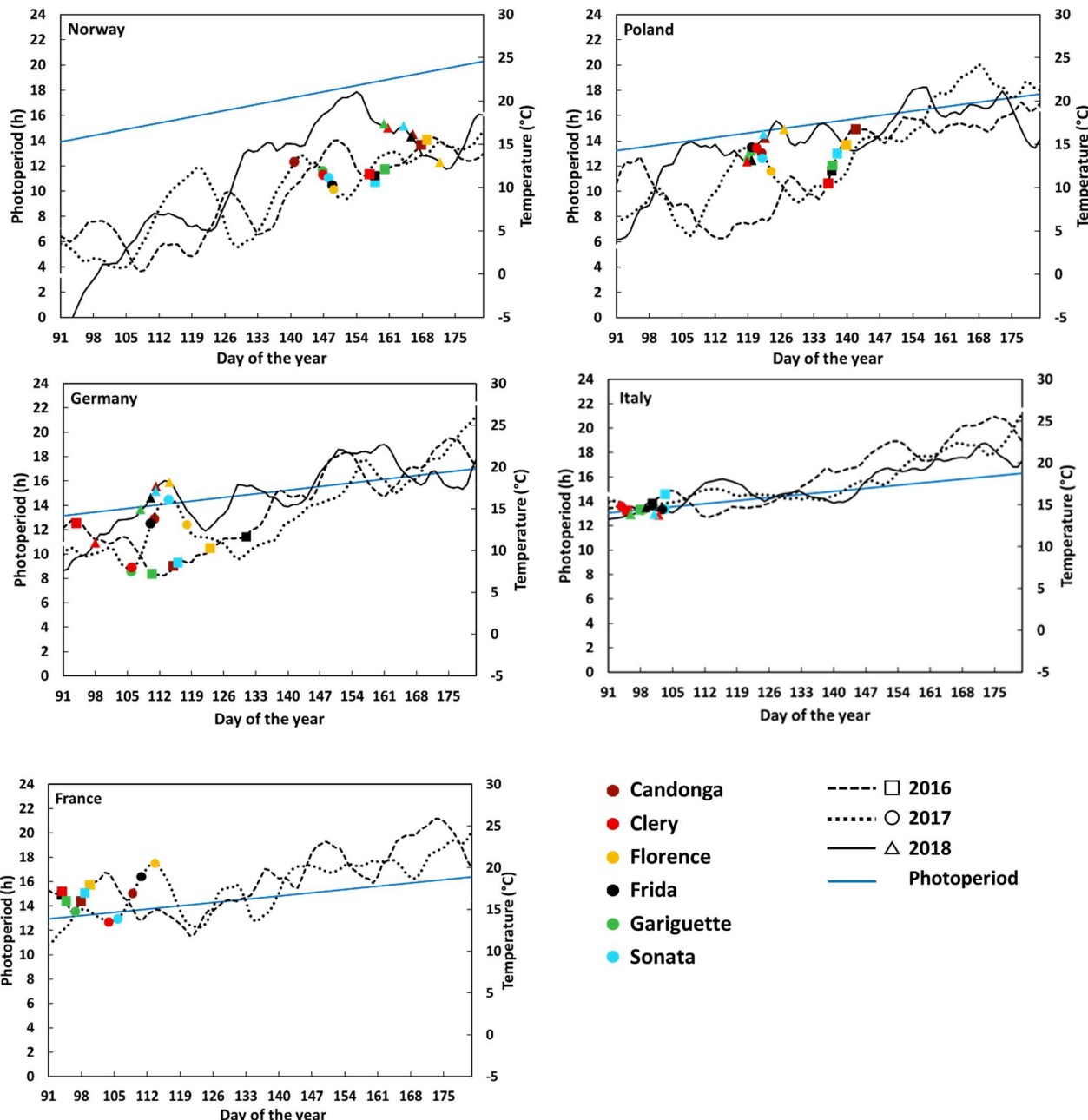

**Figure A3.** Day of anthesis of six cultivars grown at five locations in Europe as influenced by temperature and photoperiod for 3 years. Data are presented as running means of 7 days.

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
