# Peer review of "Flowering Phenology of Six Seasonal-Flowering Strawberry Cultivars in a Coordinated European Study"

_horticulturae, doi:10.3390/horticulturae8100933_

Round 1
Reviewer 1 Report
19-08-2022
Dear Editor and Authors,
The aim of the manuscript “Flowering phenology of six seasonal-flowering strawberry cultivars in a three-year coordinated transnational European study” was to study six genetically distant strawberry cultivars for three 3 years in relation to measured climatic parameters at five european locations (Norway, Poland, Germany, Italy, and France) in order to evaluate the climatic adaptation and plasticity of strawberry genotypes.
Although, at first glance the manuscript seems to be difficult to understand because it has scientific ambiguities, omissions and stilted language, I believe that the future potential scientific reader of the paper will find interesting the insight into the climatic adaptation and plasticity of strawberry genotypes.
Authors present a quite interesting topic, but the manuscript must be improved further in order to be published in the Journal. Aiming to help authors to improve the manuscript and be published by the Journal, I hereby send you the 17 most important corrections/suggestions for the manuscript in order to revise it for publication.
So, my Overall Recommendation for the manuscript is:
“Accept after minor revision (corrections to minor methodological errors and text editing)”.
Kind regards
The Reviewer #
=============================================================
The 17 most important comments/suggestions for your manuscript are:
1. Authors should read carefully the “Instructions for Authors” of Horticulturae MDPI Journal and especially the “Materials and Methods” section.
2. Authors should revise the abstract properly in good, grammatically correct English. The methodology of the abstract is questionable.
I suggest authors should state the significance of the results achieved taking into consideration the specific objectives, the data they present in the manuscript and their statistical correlation results.
3. Line 130-131: Authors wrote: “Details of the respective latitude, altitude, yearly mean temperature, and cultivation conditions are given in Table 1.”
and Table 1: Authors wrote: "Table 1. Geographic location, soil and climatic conditions and cultivation type for the five experimental locations in Europe.”
Authors should take into consideration the above red highlights and I suggest that they should revise their sentence, and also, the title of Table 1 properly for scientific terminology and in good, grammatically correct English.
The yearly mean temperature is only one parameter of the climatic conditions, it does not represent the whole climatic conditions at the five European locations.
Also, why authors did not write in Table 1 title “yearly mean temperature” as they stated in lines 130-131 that this information is given in Table 1?
In lines 130-131 they state “…and cultivation conditions are given in Table 1.” and in Table 1 wrote “..cultivation type…”.
Authors should use the same terminology and should change in line 131 the “…cultivation conditions …” to “…cultivation type …”.
4. Line 123-148, Section 1: Authors wrote: “2.1. Experimental sites, plant material and cultivation”.
Authors in 2.1 section (page 3) of “Materials and Methods” do not give any information about the cultivation seasons… The potential reader must read till the end of page 6 (line 209) in Results, in order to find out that the experimental years were 2016 – 2018.
Authors should take into consideration the above red highlights and I suggest that they should revise the section properly writing in “Materials and Methods” clearly that the experimental years were 2016 – 2018, beginning in August and ending in… for scientific soundness using third person writing and good, grammatically correct English.
5. Line 133-134, Table 1: Authors present in Table 1 “Yearly mean temperature (°C)a” for “a For the period 1981 - 2010”.
Since, authors performed the experiments between 2016-2018, I suggest that they should revise Table 1 and present Yearly mean temperature data for the period 1981 – 2018.
Moreover, in order to give readers a clear insight into climatic conditions at the five European locations used, authors should present also in Table 1:
The Yearly mean temperature for the period 2016 – 2018,
The Yearly mean temperature for the year 2016,
The Yearly mean temperature for the year 2017,
The Yearly mean temperature for the year 2018,
The Yearly mean global radiation for the period 2016 – 2018,
The Yearly mean global radiation for the year 2016,
The Yearly mean global radiation for the year 2017,
The Yearly mean global radiation for the year 2018,
The Yearly mean Rainfall for the period 2016 – 2018,
The Yearly mean Rainfall for the year 2016,
The Yearly mean Rainfall for the year 2017,
The Yearly mean Rainfall for the year 2018.
The Yearly mean Irrigation amount for the period 2016 – 2018,
The Yearly mean Irrigation amount for the year 2016,
The Yearly mean Irrigation amount for the year 2017,
The Yearly mean Irrigation amount for the year 2018.
6. Line 131-132: Authors wrote: “Air temperature (Tmean, Tmax and Tmin) and global radiation were recorded at each location.”.
and in Line 188-189: Authors wrote: “In addition, global radiation was also calculated for the 3-week period before anthesis.”
The manuscript seems to be difficult to understand because it has scientific ambiguities and omissions.
What about the global radiation?
It was recorded (meaning measured) as stated in lines 131-132 or calculated for the five locations as stated in lines 188-189?
or perhaps the mean global radiation was calculated for the 3-week period?
or perhaps the sum global radiation was calculated for the 3-week period?
The future potential reader of the Journal “Horticulturae: should deal with a clear scientific article without ambiguities and omissions.
Authors should revise the manuscript.
7. Line 146-148: Authors wrote: “Plant protection, fertilization and irrigation in open field sites, and fertigation of the plants in peat bags were performed according to local guidelines.”
Authors should take into consideration the above red highlights and I suggest that they should revise the section 2.1 properly for scientific soundness.
They are writing “…were performed according to local guidelines.”
Local guidelines by whom?
By farmers, by a scientific institution (and which), by an agronomist, by a third person…?
Authors should give a brief description of plant protection, fertilization and irrigation in open field sites, and fertigation of the plants in peat bags.
These parameters and especially irrigation water, possible water stress and nutricients could have a strong effect on final results. Authors should have statistically checked for possible effects.
Besides authors themselves wrote in “Introduction”:
“In addition to photoperiod and temperature, other environmental factors such as water, mineral nutrients and radiation can also influence the flowering process in strawberry.
A complication with these factors is, however, that they may influence flowering both directly and indirectly through their effect on growth. Thus, temporary drought and nutrient deficiency stress may cause floral initiation under otherwise non-inductive conditions [2,3], whereas at the same time irrigation and fertilization may increase flowering by producing a larger plant with more potential flowering sites.”
8. Line 151-152: Authors wrote: “At each sampling date, the main crown of four plants from each of three replicates (i.e., 12 crowns of each cultivar) were…”
Authors should take into consideration the above red highlights and I suggest that they should revise their section properly for scientific soundness. So far, authors mention three replicates, but they did not mention the treatments clearly…!
9. Line 166-167: Authors wrote: “Here, we use floral Stage 2 for day-of-the-year of flower initiation (DOY-FI). DOY-FI was interpolated using the spline smoothing function.”.
Although, in “Materials and Methods” authors wrote that they performed flower initiation interpolation using the spline smoothing function, in the “Results” section they presented the diagrams with the interpolation without mentioning anything about the interpolation errors which is a vital quantitate outcome of the interpolation.
I suggest that they should revise their manuscript and present also the interpolation error for each Cultivar.
10. Line 193-206:
Authors in section “2.5. Statistical analyses” they do not present their zero hypothesis.
They should take into consideration the above and I suggest that they should revise the 2.5 section and present clearly their zero hypothesis.
11. Line 207 and beyond:
The manuscript seems to be difficult to understand because it has scientific ambiguities and omissions.
Authors should present a Table with the descriptive statistics (minimum, maximum, mean, st. deviation, variance, Skewness and Kurtosis) of the climatic parameters and the results of each cultivar and each location for flower initiation.
Moreover, it would be very helpful to include in the Table, the statistics of rainfall at each location, the irrigation water amount supplied at each location and each cultivar, and the nutrients supplied at each location and each cultivar.
12. Line 212: Authors wrote: “…statistical interpretation are presented in Figure 1 and Appendix A, Table 1.”.
Authors should take into consideration the above red highlights and I suggest that they should revise as Table A1.
13. Line 271: Authors wrote: “Appendix A, Table 2 demonstrates differences between both cultivars …”.
Authors should take into consideration the above red highlights and I suggest that they should revise as Table A2.
14. Figure 2, 4, 6 and 8:
Figures 2, 4, 6 and 8 are of low quality.
Authors should revise them and improve their quality.
Moreover, in these figures, in loading plots (b), the climatic variable captions are very close and confusing. It would be better if an arrow was used for each climatic variable caption pointing to the corresponding line of the loading plot.
15. Line 561-562, Figure A5:
Authors should revise Figure A5.
The diagrams of Norway and Poland have only the half upper part of the X axis title. The other half is missing.
Authors should revise them.
16. In Results for the PCA analyses:
In each PCA (factor analysis) did authors used the R-mode factor analysis for extracting factors or not?
How many components (factors) did you find for each PCA (factor analysis)?
What was the optimal number of components (factors) for each PCA (factor analysis)?
If all the factors for each PCA (factor analysis) taken together what percentage explain of the total variance in each data matrix?
Authors should revise the results and answer the above questions.
17. In Results the presentation of statistical and PCA analyses:
Authors should review, try not to present only a good outcome of the analysis, revise and present more carefully their analysis and explanation.
Example:
Authors wrote: “Another PCA analysis performed to further assess the effect of environment during the first 5 weeks after FI on number of crowns and flowers (Figure 6), revealed that the climatic conditions during this five week-period had a stronger effect on number of flowers than on crowns. These results were confirmed by the detailed analysis performed using linear regression (Figure 7a, b).”
An example of transparent analysis would be as follow:
In figure 7.a, the Global radiation [W m-2]) during the five week-period and the number of flowers resulted in low to very low coefficients of determination for the six cultivars, which are classified in descending order as ‘Gariguette’ (FR), ‘Florence’ (UK), ‘Clery’ (IT), ‘Sonata’ (NL), ‘Candonga’® (ES) and ‘Frida’ (NO). Results for cultivars indicate that 4.9-38.1% of the variation in the outcome has been explained just by predicting the outcome using the covariates Global radiation and number of flowers. The model partially predicts the outcome for the six cultivars.
In figure 7.b, the Global radiation [W m-2] during the five week-period and the number of crowns resulted in very low to zero coefficients of determination for the six cultivars which are classified in descending order as ‘Frida’ (NO), ‘Sonata’ (NL), ‘Clery’ (IT), ‘Florence’ (UK) and ‘Gariguette’ (FR) and ‘Candonga’® (ES) both at the last class. Results for cultivars indicate that 0.0-9.3% of the variation in the outcome has been explained just by predicting the outcome using the covariates Global radiation and number of crowns. The model does not predict the outcome for ‘Gariguette’ (FR) and ‘Candonga’® (ES). The model partially predicts the outcome for ‘Frida’ (NO), ‘Sonata’ (NL), ‘Clery’ (IT) and ‘Florence’ (UK).
Comparison of both analyses (Figure 7.a and 7.b), although, it revealed that the global radiation during this five week-period had a stronger effect on number of flowers than on crowns, both are resulted in low to very low percentages of variance explanation, indicating that many other underlying factors exist.
I wish you a constructive revision!
-END of REVIEW-

Author Response
Please see the attachement

Author Response
Please see the attachement

Round 2
Reviewer 2 Report
Horticulturae 1882154 Revised
Many thanks to the authors for the revised submission, which is much improved over the original manuscript. The revised paper should be suitable for publication in the Journal. I provide the following suggestions.
Indicate in the ‘Abstract’ that the relationships between flowering and crown production and weather were highly variable across the different sites and cultivars. Provide further details about the sites/cultivars where there were strong relationships, and maybe a brief explanation why this occurred.
I still feel that the paper could be improved by reducing the size of the ‘Introduction’ and ‘Discussion’.
Some of the details provided in the ‘Conclusion’ should be moved to the ‘Discussion’
The ‘Conclusion’ should just highlight the main findings of the study in no more than 3-4 sentences.
Author Response
Please see the attachement
